# The Influence of Emotion Induced by Accidents and Incidents on Pilots’ Situation Awareness

**DOI:** 10.3390/bs13030231

**Published:** 2023-03-07

**Authors:** Tianjiao Lu, Yuan Li, Chenchen Zhou, Menghan Tang, Xuqun You

**Affiliations:** 1Student Mental Health Education Center, Northwestern Polytechnical University, Xi’an 710062, China; 2Shaanxi Key Laboratory of Behavior and Cognitive Neuroscience, School of Psychology, Shaanxi Normal University, Xi’an 710062, China

**Keywords:** pilots, accidents, incidents, emotion induced, situation awareness

## Abstract

(1) Background: This study examines the differences in emotions induced by accidents and incidents as emotional stimuli and the effects on pilot situation awareness (SA) after induction. (2) Method: Forty-five jet pilots were randomly divided into three groups after which their emotions were induced using the pictures of accident, incident, and neutral stimulus, respectively. (3) Results: The conditions of accidents and incidents both induced changes in the pilots’ happiness and sadness and the changes in the emotion were regulated by the emotional intelligence of pilots in the high SA group. The emotion induction, which caused a direct change in pilot’s happiness and fear, resulted in conditions that indirectly affected level 1 of SA in pilots. (4) Conclusions: The research elucidates the difference between accident and incident in inducing pilot emotions, and reminds us that SA level exerts the regulating effects on the same emotional induction conditions.

## 1. Introduction

Rapid progress in the aviation industry, escalating pressures in airline flights, and the growing challenges posed by pilot automation have increased aviation accident and incident rates in recent years [1]. Additionally, according to research [2,3], in the aviation industry, 51.6% of accidents and 31.5% of incidents are attributed to wrong decisions, which are partly to blame for SA (situation awareness) errors. SA, as the critical factor in aviation security [4] and priority in pilot risk management [5], dramatically impacts the guarantee of aviation security. Researchers have outlined different SA definitions [6,7,8,9]. Endsley’s (1995) definition of SA has been universally acknowledged [10]. The perception of each factor in the current situation, comprehension of meaning, and prediction of the future state have all been widely adopted. Researchers led by Endsley did detailed research on pilot SA from the perspective of measurement techniques, influence factors, and training [11]. With different cognition abilities and multiple processing stages in SA formation and maintenance stage, the interplay among multitudinous influence factors, such as pilot’s state, cognitive level, task difficulty, external environment, etc., is very complicated [12,13,14,15]. Firstly, the influence of the cognitive level on pilots is visible [16]. Through rigorous screening mechanisms, research clearly shows that pilots’ cognitive level prepares them for the flight mission. Secondly, the pilot’s state, including physical and mental, is uncontrollable and unstable. These factors are susceptible to daily life, social events, and occasions with formidable volatility, either directly or indirectly, these matters should be appreciated [17]. Besides pressures and emotional influences arising from life and social events, aviation accidents have a direct effect on the pilot’s emotional state, cognitive abilities, and even the execution of missions. The existing literature ascertains that aviation accidents affect pilots’ psychological and spiritual health, and even trigger posttraumatic stress disorder (PTSD), panic disorder, and depression [18].

Though pilots pass through rigorous screening, and the accumulated flight missions help them stabilize emotions better than ordinary people, they are still vulnerable to aviation accidents (with severe casualties and airplane damage) and incidents (successful landing with minor damage to the airplane and no casualties) closely related to life. We assume that the main difference between accident and incident is the degree of damage caused. This difference may induce different kinds or levels of emotions in pilots. The varying emotional influences have not been researched before. Additionally, previous research on SA vital to aviation security focuses mostly on the influence of overall cognitive abilities on SA [13,19] and fails to address the influence of pilot emotions on SA. We assume that aviation accidents can be induced by pilots’ negative emotions or loss of SA, and possibly impact pilots’ emotions. Both SA abilities and emotional intelligence offer regulating effects.

Through examining how the accidents and incidents induce pilots’ emotions and investigating varying influences of accidents and incidents on SA in combination with the aforementioned factors, this research aims to confirm that SA formation and maintenance are under the influence of emotions in the full process and such influence is further manipulated by pilots’ emotional intelligence and mediating or regulating effects of personal SA.

### 1.1. Human Factors in Aviation Accidents and Influence on Pilot Emotions

Investigations into recent aviation accidents and incidents show that human error is the main contributing factor [20]. Flight accidents and incidents not only cause direct injuries to victims and airline companies but also negatively affect the mental state and physiological functions of other pilots in the unit. In recent decades, researchers [21,22,23] have all outlined the conspicuous negative impacts of aviation accidents on pilots’ physical and mental health. Alternatively, some researchers have examined non-involved pilots’ physical and psychological state in the unit after the occurrence of accidents [18,24,25]. The above findings illustrate how aviation accidents and incidents aggravate pilots’ depression, compulsion, anxiety, fear, and other negative emotions. It is clear that aviation accidents and incidents inevitably affect pilots’ emotional states. In the aviation domain, an accident signifies more severe affairs with casualties, while an incident signifies a case without severe injuries [26]. Though both accidents and incidents exacerbate the psychological burdens and emotional volatility of pilots, the research assumes a discrepancy between accidents and incidents with varying severity in emotional induction potentials. 

### 1.2. Influence of Pilot Emotion on Aviation Security and SA

This research focuses on pilot emotions because pilots’ emotional state directly or indirectly affects their physical health, mental health, and cognitive abilities, which further endangers their SA when undertaking flight missions and endanger aviation security. Existing research shows that pilots’ emotional problems can directly induce aviation security accidents [27]. Specifically, pilots’ physical and mental state dramatically restrains their efficiency and validity of information processing [28]. When they get trapped in a poor physical and mental state, most pilots generally lose their ability to react fast and skills to process significant amounts of information. SA, being essential to flight missions, is sustained by pilots through the real-time perception of sundry information in the present environment. There is no related study on the influence of the emotional state on SA. This derives the hypothesis that emotional states have an impact on SA, and various impacts of different kinds of emotions.

### 1.3. Influence of Emotions on Cognitive Abilities and Regulating Effects of Emotional Intelligence

Firstly, the existing literature has revealed the importance of emotions to SA and pilot emotions. Secondly, this paper does not overlook the intimate contact between emotions and cognition abilities. Though the impact of emotions on SA has been significantly highlighted [29], the relation between the two has not been researched. As a result of the close contact and immediate impact between emotions, thinking, decisions, and actions, this paper assumes that the influence of emotions on SA is inexorable. Emotions reflect, affect, and deviate from cognition to the meantime and control choices in the cognitive process. Firstly, the “feeling as information” theory proposed by Schwarz and Clore [30] outlines those emotions can explicitly affect judgment. Secondly, the emotional model of Loewenstein and Lerner [31] highlights that different from expected emotions and subject to cognitive assessment, immediate emotions (i.e., immediate visceral reaction) offer direct effects on decision-making and cognitive assessment. Generally, pilots’ emotional states may explicitly affect SA comprehension and prediction. According to Lerner and Keltner [32], people are apt to extract and process information compliant with present emotions. This shows that individuals with positive emotions have happy experiences and better judgment [33], while individuals with negative emotions portray sad memories and have effects of cognitive assessment [34]. According to this research, the ability to percept and control emotions plays a mediating role in the relation between emotions and SA and also offers the role of pilot’s emotional intelligence. Emotional intelligence refers to the ability of individuals to identify and understand their own and others emotional states, and to use this information to solve problems and adjust their behaviors [35]. It is related to an individual’s ability to understand, control, and regulate emotions. There are related studies that show that emotional intelligence is positively correlated with professional competence and task performance [36].

By integrating the influence of aviation accidents on emotions and SA, this research paper outlines four hypotheses below: 

**H1:** *Accidents and incidents exert varying influences on pilots’ emotional induction*.

**H2:** 
*Emotional intelligence mediates the effects of accidents and incidents on pilots’ emotions and baseline SA level (before induction).*


**H3:** *Accidents and incidents and post-induction emotional changes follow different functional mechanisms on pilots’ post-SA emotional induction*.

**H4:** *Accidents and incidents’ influence on pilots’ post-induction emotions, self-regulation, and post-induction SA is regulated by pilot SA level and induction conditions*.

## 2. Materials and Methods

### 2.1. Participants

Based on previous pilot application studies [37,38,39,40,41], the number of subjects generally used is about 6–16. Therefore, we selected 45 subjects for this study. Male jet pilots (*n* = 45) were sampled with an age range of 25 to 45 years, 12 people aged 25~30 years (26.66%), 33 people aged 31~45 years (73.33%), with an average of 1510 flight hours (*SD* = 898) took part in this study. They were randomly assigned to three equal groups (accident group, incident group, neutral group). This research complied with the China Psychological Association Code of Ethics and was approved by the Institutional Review Board at Shaanxi Normal University. Informed consent was obtained from each participant.

### 2.2. Materials

Emotion Elicitation: Firstly, accident-related pictures were chosen according to the different characteristics of aviation accidents and incidents [26]. Thirty accident pictures with severe casualties and airplane damage as material for emotion elicitation were selected. Thirty pictures of successful landing with minor damage to the airplane and no casualties, as material for the emotion elicitation of the incident, were also selected. The other 30 pictures with safe flight as medium stimuli were selected (Figure 1).

Emotional rating: The emotional state and the emotional evaluation of the pictures were taken using the 7-point Likert-type scale of numbers between each emotional dimension as the score of the participants’ emotional state. One meant that there was no certain emotion at all, 7 meant that it was full of a certain emotion, and a score from 1 to 7 indicated that the degree of emotion gradually increased. The emotional dimensions included: Happiness, Anxiety, Calmness, Disgust, Surprise, Anger, Fear, Sadness.

In order to confirm the effect of stimulation on emotion, a pilot study was conducted before the experiment. We invited 30 professional pilots to randomly evaluate the emotions of each picture in neutral, incident, and accident group. The emotional dimensions of each picture were rated using the 7-point Likert-type scale. The evaluation results showed that the pilots’ evaluation of the induced pictures was based on picture groupings. The variance for the evaluation of each picture showed significant differences among the three groups (*F* (2, 83) = 440.25, *p* < 0.001), the degree of negative emotion was greater in the accident group (*M* = 6.41, *SD* = 0.54), the emotional evaluation of the incident group was neutral (*M* = 4.39, *SD* = 0.45), and the neutral group was the most positive (*M* = 2.05, *SD* = 1.79).

In the formal experiment, each pilot assessed his current emotional state at different dimensions (7-point Likert-type scale) before and after the emotion elicitation task.

During the emotion elicitation part, each picture used 10 s, and the pilots evaluated their emotions at different dimensions (7-point Likert-type scale) based on each picture. 

SA measurement: To achieve the repetition of the subjective evaluation and enrich the results of the study, we used the two most widely recognized SA measurement methods, SART and SAGAT, to conduct the pretest and posttest, respectively. Before the induction, the SART was used for subjective assessment of pilots’ SA. After induction, emotions affected subjective evaluation, and the use of the simulated flight measurement SAGAT caused the emotion-induced state to subside. As a result, we referred to Sohn and Doane (2004) experimental program to measure SA based on SAGAT, which was achieved by taking screenshots of flashing flight cross-section, simulating technical questions of SAGAT freezing, and objectively measuring the SA of the pilots [13].

Situation awareness rating technique (SART) is a simplistic post-trial subjective rating technique originally developed for the assessment of pilot SA [9]. SART uses the following ten dimensions to measure operator SA: Familiarity with the situation, focusing of attention, information quantity, information quality, instability of the situation, concentration of attention, complexity of the situation, variability of the situation, arousal, and spare mental capacity (Figure 2).

SAGAT test: Similar SA task was set by referring to Sohn and Doane [13] and simulating the SAGAT freezing technique (question content, number of questions, and the scoring methods) (Figure 3). The pilots were required to answer questions from three levels, with four questions in each level (the program example is shown in Figure 3). The result of SAGAT was the correctness of the questions in each level recorded by the program, and the total score was added to a sum that represented the quality of each level.

Emotional Intelligence Scale: Wong and Law Emotional Intelligence Scale (WLEIS) [42]. The WLEIS is a 16-item self-report trait EI measure using a 5-point Likert-type scale (1 = totally disagree to 5 = totally agree). The measure consists of 4 items: Self-Emotion Appraisal (SEA), Other’s Emotion Appraisal (OEA), Use of Emotion (UOE), and Regulation of Emotion (ROE).

### 2.3. Procedure

Firstly, all three groups of participants completed SART and WLEIS before the experiment. The pilots were asked to close their eyes and calm down with a short period of rest. Then, the pilots were required to evaluate their current emotional state subjectively. The three groups of participants were divided into accident group, incident group, and control group (neutral stimulation) and watched corresponding picture stimuli, respectively. The emotional elicitation task was conducted by E-prime software, and the presentation time of each picture was 10 s. After the emotional elicitation task was completed, the pilots were asked to make a subjective evaluation of their current emotional state again. Then the pilots undertook the SAGAT test immediately after the evaluation (Figure 4). 

### 2.4. Data Analyses

All statistical evaluations were done using Statistical Package for Social Sciences (SPSS) version 22.0. The independent *T*-test was applied to test the changes in emotion after the induction and evaluation of the results of stimulus pictures. Relationships among induced groups, emotional intelligence, and induced emotional changes in all dimensions were established using a general linear model test. The SART scores were used to group high and low score groups and compare the impact on emotion changes in different dimensions and emotional intelligence under the condition of different induction factors. The stepwise regression equation model was applied to establish the degree of induction, SART, and the influence of emotion changes on the SAGAT results after induction. Followed by that, we gradually specified the mutual influence between them, and then a conditional process model was established. Among them, we used the score of each emotion dimension before changing to divide the score of that after induction to calculate the change rate of emotion.

## 3. Results

### 3.1. Pilot’s Emotional State of Various Dimensions under Different Induction Conditions

Using the ratio of the difference in the grade evaluation of each emotional dimension before and after the induction to the original emotional evaluation level as the emotional change rate, the differences in the emotional change rates of the pilots were compared under different induction conditions (Figure 5). Variance test results found significant differences among three evoked conditions of happiness (*F* (2, 42) = 14.42, *p* < 0.001, *η*^2^ = 0.41, *post-hoc power* = 0.94) and sadness (*F* (2, 42) = 7.07, *p* < 0.01, *η*^2^ = 0.25, *post-hoc power* = 0.78). LSD-comparison among all the happiness emotions, a significant difference was observed among neutral (*p* < 0.001), accident group, and incident group (*p* < 0.001). Similar results were found for sadness emotions (*p* < 0.01). For other emotions evaluated, there was no significant difference among the three groups (*p* > 0.05).

### 3.2. Differences of Emotional Intelligence between Groups under Clustering of SART

The results of K-means clustering analysis conducted on the four scoring types of SART, divided the subjects into high and low groups (Figure 6). A *T*-test performed on the two groups’ SART scores found significant differences in the scores of demands (*t* (43) = −4.89, *p* < 0.001, *d* = −1.46, *post-hoc power* = 0.90), supply (*t* (43) = 4.40, *p* < 0.001, *d* = 1.32, *post-hoc power* = 0.80) and understanding groups (*t* (43) = 4.00, *p* < 0.001, *d* =1.20, *post-hoc power* = 0.68), SART (*t* (43) = 8.32, *p* < 0.001, *d* = 2.47, *post-hoc power* = 0.99). The *T*-test of WLEIS found no significant differences in the scores of various dimensions of emotional intelligence (SEA, OEA, UOE, ROE, and total score) in the groups of high and low SART score (*t*_1_ (42) = 1.33, *p*_1_ = 0.19; *t*_2_ (42) = 0.94, *p*_2_ = 0.36; *t*_3_ (42) = 1.21, *p*_3_ = 0.23; *t*_4_ (42) = 0.38, *p*_4_ = 0.71; *t*_5_ (42) = 0.94, *p*_5_ = 0.35).

### 3.3. The Effect of Pilot’s Emotional Intelligence on the Rate of Emotional Change after Induction

By dimension reduction analysis of the four dimensions of WLIES, we obtained a comprehensive index representing the level of comprehensive emotional intelligence. General linear model results revealed the change rate of each emotional dimension to be directly and indirectly affected by emotional intelligence and evoked conditions, as shown in Table 1.

Comparison of happiness in neutral-induced control group shows a significant emotion change in pilots in the high-score group under both accident and incident conditions (*R*^2^ = 0.47). Similarly, the low-score group had a significant change under the negative conditions (*R*^2^ = 0.42). For anxiety, pilots in the high-score group show a general emotion fluctuation under the induction of accident condition (*R*^2^ = 0.37), however, emotion change is observed in the low-score group (*R*^2^ = 0.15). For calmness, the effects of emotional intelligence were significant in the high-score group (*R*^2^ = 0.52). The interaction between accident condition and emotional intelligence was also found to be significant, but the impact on the low group is not significant (*R*^2^ = 0.45). In terms of aversion, pilots in high-score groups (*R*^2^ = 0.68) exhibited significant changes under the condition of incident and on interaction with emotional intelligence. Pilots in the low-score groups (*R*^2^ = 0.19) also show significant changes under the condition of accident, the main effect of emotional intelligence and interaction effect.

As for surprise emotion, pilots in high-score group (*R*^2^ = 0.35) reported significant changes under the condition of accident, along with the main effect of emotion intelligence. At the same time, there was a significant main effect found in the low-score group (*R*^2^ = 0.15) under the two inductions of emotional intelligence. No significant changes were observed in both groups when anger was assessed (*R*_1_^2^ = 0.28, *R*_2_^2^ = 0.11). For fear emotion, there was a significant main effect of emotional intelligence in the high-score group (*R*^2^ = 0.41), and the interaction between accident condition and emotional intelligence was also obvious. Each factor in the low-score group (*R*^2^ = 0.17) was not significant. In terms of sadness emotion, high-score group (*R*^2^ = 0.54) had significant changes under the condition of incident, the main effect of emotional intelligence was also significant, and the interaction effect of accident condition and emotional intelligence was significant as well. However, the low-score group (*R*^2^ = 0.26) only had significant changes under the condition of accident.

### 3.4. The Effects of SART and the Rate of Emotional Change on SAGAT after Induction

We used SART as control variables to compare the influence on SAGAT after emotion changes under the condition of two evoking factors. The results of the hierarchical stepwise regression equation are shown (Table 2). Under the condition of accident induction, the significant influence of emotion changes on level 1 was found (*R*^2^ = 0.45, Adjust *R*^2^ = 0.41, *F* (1, 14) = 11.42, *p* = 0.004), and it also explained 41% changes of level 1. In terms of level 2, demand is obvious under the condition of incident (*R*^2^ = 0.30, Adjust *R*^2^ = 0.25, *F* (1, 13) = 5.67, *p* = 0.033). As for level 3, the model of sadness changes induced by accident was significant (*R*^2^ = 0.30, Adjust *R*^2^ = 0.25, *F* (1, 14) = 5.93, *p* = 0.029). In terms of SAGAT, sadness was still significant when induced by accident (*R*^2^ = 0.35, Adjust *R*^2^ = 0.308, *F* (1, 14) = 7.67, *p* = 0.015), as well as demand (*R*^2^ = 0.31, Adjust *R*^2^ = 0.26, *F* (1, 13) = 5.79, *p* = 0.032).

### 3.5. A Conditional Process Model of Evoked Pilots’ SA: Mediating Effect of Emotion and Moderating Effect of Emotional Intelligence

We used each emotional dimension as an independent variable, SAGAT as the dependent variable, and other factors as moderating or mediating variables. By using the Hayes Process v3.0 with the bootstrapping strategy (*N* = 5000), the following conditional process model (Figure 7) was finally obtained, which explained the emotional intelligence, the evoked conditions, the emotional changes, the pre-evoked pilot SA’s effect on the SAGAT after induction, and the complex processes [43]. The main findings are summarized in Table 3 and Table 4.

According to the conditional process model proposed by Hayes [43], in the model of emotional intelligence → calm → anger → SAGAT, changes in calm and anger emotions play a chain-mediating role in emotional intelligence and SAGAT, which is regulated by the induced conditions. Under the condition of accident, the equation tests the parameters index = 0.07, *95% CI* = [−0.10, 0.18], and under the incident conditions, the equation tests the parameters index = −0.13, *95% CI* = [−0.26, −0.01]. Among them, emotional intelligence only has a direct influence on calm emotion. Moreover, the process is also regulated by the attention and information processing ability of SART. Furthermore, the changes in calm emotion also influence SAGAT by mediating the effects of anger. In this process, the regulating effect of the induction condition was significant, and this effect was only significant under the induction of incident. Finally, demand before induction was viewed as a covariate, which had a significant direct effect on SAGAT after induction.

## 4. Discussion

This research takes accidents and incidents as emotional induction conditions to compare pilots’ emotional induction. The two conditions both successfully induce changes in pilot emotions. The direct discrepancy of induction emotions under two conditions is less significant because the pilots’ emotional changes remain stable within a range of stimulus intensity. Additionally, pilots’ personal flight experience is also at play. Professional pilots cope better with emotional stimuli and ensure emotional stability. SA high-grouping emotional intelligence, respectively, generates direct and indirect influence on some emotional changes. Eventually, under accident and incident conditions, induced emotional changes differ in dimensions and degrees, and likewise, follow-up emotions also differ in the influence on pilots’ post-induction SA. Pilot SART demand causes a direct influence, for SART places emphasis on the regulating effects of information processing abilities.

### 4.1. The Difference in Pilots’ Emotional Change Rate Is Insignificant under Accident and Incident Induction Conditions 

This is against Hypothesis 1 that accidents and incidents exert varying influences on pilots’ emotional induction. Regardless of the validity of stimulus materials (verified), we ascribe it to senior pilots’ high emotional stability [44] and inadequate stimulus distinction between accidents and incidents. Secondly, the two induction conditions preference of top-down emotional induction process gets the sophisticated role of cognitive process and emotional regulation involved. Given the professionalism and situation meaning of aviation accidents as a kind of induction material, top-down emotional induction includes the cognitive process. Individuals’ emotional reaction is dynamic beyond the restraints imposed by perception and stimulus [45]. Therefore, this research acknowledges the relation to pilot experience. Based on existing participants’ SA grouping, we find under accident and incident induction conditions, there indeed exists some discrepancy among pilots in high and low-score groups in emotional change rates at different dimensions. Moreover, because of the insignificant differences among pilots in high and low-score groups in emotional intelligence, we may deny the direct role of emotional intelligence difference and notice the critical role of SA grouping. The finding tests the validity of Hypothesis 1, but simultaneously requests the addition of one more control factor, pilot SA (or experience). Accident and incident conditions indeed show a difference in pilots’ emotional induction degree and dimension, in which SA plays regulating effects in this process. Though empirical differences have not been investigated in the research, flight experience is generally positively correlated with the SA level [16]. We now predict that such difference will be more conspicuous in newbie and expert grouping comparisons. Specific conclusions are about to be verified in further research.

### 4.2. Emotional Intelligence Mediates the Effects of Accidents and Incidents’s on Pilot Emotions and Baseline SA Level

Diverse emotions are experienced subject to different induction conditions and emotional intelligence. This diversity is also influenced by the sophisticated interplay among all factors. On the premise of inconspicuous emotional intelligence, the difference between high and low-score SA groups is happiness and sorrow. The low-score SA group is only directly affected by accident induction conditions. On the other hand, higher SA scores suggest more apparent emotional changes among pilots. Only happiness and anxiety are directly influenced by induction conditions. Other emotional changes, such as aversion, fear, sorrow, and serenity, are influenced by the interplay between emotional intelligence and induction conditions. These findings were contradictory to those of a study done by Goeters et al. [46], which reported that high SA has a positive correlation with empirical skills. The study further reported that a higher pilot performance and SA corresponds to higher emotional stability. Based on different emotional dimensions and the significance of cognitive actions, we suggest that these results completely represent pilots’ perception of an emotional stimulus, emotional volatility, and recovery of the initial emotional state. This process is likely associated with top-down emotional induction. On the premise of emotional intelligence participation, pilots in the high-score SA group do not always maintain their emotions stable after an emotional stimulus. This involves perception of the stimulus as well as assessment of the cognitive process. Accident and incident stimulus induction elicits more cognitive meaning and regulates emotional changes even if there is no difference in the subjective assessment of emotional perception in high and low-score SA groups. This coincides with the top-down emotional stimulus properties [47]. Based on these findings, emotional intelligence plays a different role in the regulation of emotional changes under two induction conditions. The low-score SA group experienced significant changes in levels of happiness and sorrow when directly subjected to an accident. No regulating effects of emotional intelligence were involved. Pilots in the high-score SA group significantly suffered from the regulating effects of emotional intelligence on emotional volatility, except for happiness and anxiety. This result focused on the peculiar differences in different emotional dimensions [48]. When individuals are conscious of a stimulus of ecological significance, basic emotions, such as happiness, sorrow, fear, anger, and will, apace automatically and unconsciously come into play [49]. From the perspective of evolutionary psychology, they are biological and social functions. For instance, sorrow motivates empathy and altruistic actions [50], disgust motivates avoidance of contamination [51], while anger motivates defensive mechanisms [52] and self-protection actions [53]. Happiness and sorrowful emotions changed in the low-score SA group. They are both low-motivation emotions that arise from pilots’ perceptions and experiences of their personal emotions as well as those of others. They tend to view accidents as an emotional motivation and thus place a higher emphasis on their perception of accidents [54] instead of any superior actions and motivations. Consequently, emotional intelligence did not regulate emotional changes of happiness and anxiety (both low-motivation emotions) in the high-score group when they concentrated on their personal experiences. Along the same line, amid emotional changes of aversion and fear, emotional intelligence lowers emotional volatility under accident conditions in the high-score SA group. As indicated by former studies, aversion and fear are both high-induction and high-motivation emotions. The two emotions induce follow-up actions because of the high motivation. This phenomenon is attributed to the high-SA pilots’ higher sensitivity to the auto-self-protection mechanism. The high-score group concentrates on the “accident” properties of stimulus as well as the guiding significance of cognitive actions [55]. Further to this, it could be judged that they have a higher perception and vigilance towards environmental changes. SA is a process structure that includes access and maintenance of information state in the entire mission. This is possibly what makes them maintain high SA. 

### 4.3. Accidents and Incidents and Post-Induction Emotional Changes Follow Different Functional Mechanisms on Pilots’ Post-SA Emotional Induction

Under incident conditions, emotional volatility did not produce any influence on post-induction SA. However, pilots’ prior-induction perception of the flight situation and requirements of cognitive resources influenced their comprehension and SA because emotional volatility does not produce direct influence on post-induction SA [56]. However, the results implicitly show that pilots’ emotional regulation strategy devotes some cognition resources under incident induction conditions. No auto-regulation strategy is involved in this case. Pilot emotions show great stability and strong anti-interference performance, which is evident because of the prominent influence of incidents on general comprehension. Despite this, it is assumed that the compensation action of flight experience counteracts the influence of incident emotional stimulus on information perception and primary processing. Endsley and Garland in 2000 reported that the comprehension stage is more related to top-down information perception and bottom-up experience and knowledge combination seems to be the stage that devoted most resources and suffered significant resource competition pressures [11]. As a result, post-induction SA is lowered if pre-induction self-assessment devotes more resources and pays more effort to maintain SA. These findings are consistent with the competitive theory of emotional regulation on cognition resources [57] and outline the importance of emotional volatility and emotional regulation training of pilots. 

Emotions were found to directly affect the perception and decision-making stage of the pilots during accident conditions. This included the perception process of information (mostly bottom-up) and the process of prediction (mostly up-bottom). When intense emotions were induced, surprise generated outstanding influence on information perception and access. It also greatly affected the SA. These results were consistent with those reported by Merk and Roessingh in 2013 [58], we concluded that surprise has a great influence on information perception. According to Rivera et al. [59], surprise emotions interrupt ongoing cognitive processes but promote pilots’ vigilance and sensitivity towards current environment information. It motivates pilots to develop eminent judgment and better SA control abilities. On the other hand, sorrow directly affected the pilots’ SA owing to its immediacy and intensity. Sorrow is a feeling of strong sadness that results in emotional self-anxiety [60,61], impaired vigilance [62], and a long-term lack of attention and attention control [63,64]. The higher the sorrow change rate, the stronger the post-induction SA. This means that, in a stable emotional state, the higher the induced sorrow, the stronger the pilots’ emotional perception. This is emotional intelligence and expression of emotional sensitivity. The positive correlation between emotional sensitivity and stimulus perception possibly makes pilots have a greater sensitivity towards environmental information. That is in direct proportion to level 1. On the other hand, sorrow characterized by low motivation would not reinforce pilots’ intention and behavioral motivation but arousing individuals’ uncertainties in situation control. This would force individuals to try means to decrease such uncertainties and regain control of the situation [65]. As such, sorrow causes pilots to pay more effort and devote more cognitive resources to cut down uncertainties and restore the sense of control. Based on these findings, greater sorrow emotional volatility predicts senior cognition-orientation level 3 and general SA.

### 4.4. Accidents and Incidents’ Influence on Pilots’ Post-Induction Emotions, Self-Regulation, and Post-Induction SA Is Regulated by Pilot SA Level and Induction Conditions

To clarify the types of emotional intelligence, emotions, direct and indirect roles of SA and post-induction SA, we established an intermediary and regulating model for all factors. Indignation changes induced by incidents directly affected post-induction pilot SA. In reality, this is also restricted by pilots’ cognitive resources. This finding confirmed our projection about pilots’ emotional induction, regulation, and restoration. The interplay between information access, comprehension ability, and emotional intelligence in SART occurs in the low-score SA group. This interplay alters the way in which emotional intelligence and SA affect serenity changes. The lower the emotional intelligence, the lower the serenity degree. In this study, this phenomenon is explained based on the perception of emotional stimuli and comprehension of environmental information. It is also obvious that though emotional regulating ability plays a role in this phenomenon, it never directly acts on indignation. Instead, it undermines the volatility of other emotions by restoring the initial serenity state, which lowers indignation and negatively influences the pilot’s SA. Similarly, the mediating role only occurs among pilots with low SA in incident induction conditions. Though the regulation and transmission process of emotions conforms to the view of Kochanska et al. [66], who agree that regulating high-intensity emotions needs concentrated focus to restore the serenity state, it collides with the role of emotional regulation. It further confirms that pilots who need to concentrate on the regulation of emotions must devote their cognitive resources. This also explains the immediate role of SA demand scores in SART on post-induction SA in the model. Pilots with higher SA demand scores access fewer cognitive resources. Emotional regulation further deprives them of their limited cognitive resources. As such, resource competition that begins after the induction of emotions decreases the pilot’s SA [29].

Further analysis of the regulating effects of induction conditions revealed that even incidents without severe injuries could also cause pilots’ indignation. Contrary to existing conclusions, under the induction conditions of incidents with higher occurrence rates, pilots with low SA level were still significantly affected despite the regulating role of emotional intelligence. By contrast, accident shows more intense and explicit induction to emotions as well as more top-down perceptions. Low-score SA group leaned toward perception rather than emotional regulation. Regression analysis results proved that the high-score SA group failed to maintain post-induction SA stability.

## 5. Conclusions

Evidently, this study elucidates the importance of developing emotional regulation and reaction training for pilots to promote their SA and performance as well as aviation security. Furthermore, pertinent emotional reaction training would consolidate pilots’ defense cognitive abilities and their resistance to negative emotions in flight missions. From the perspective of application, this research analyzes the influence of accident and incident stimulus on pilot emotions and SA and, on this basis, offers the theoretical foundation for the relation between emotions and SA. However, the research still has some limitations. Firstly, the grouping of experiential factors overlooked in the paper should be taken as a control factor in follow-up research. Secondly, research results shall be further examined by more emotional induction approaches and measurement approaches. Thirdly, more SA measurement approaches, such as eye tracking, in a real flight of analog machines are advised to be taken to elaborate research results.

## Figures and Tables

**Figure 1 behavsci-13-00231-f001:**
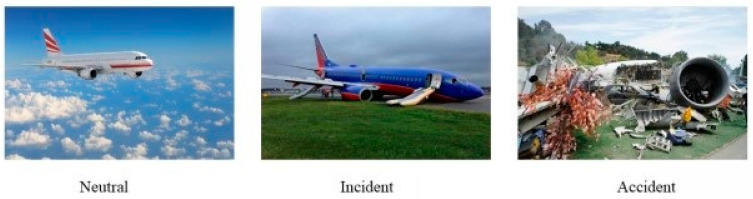
An example of materials for emotion elicitation.

**Figure 2 behavsci-13-00231-f002:**
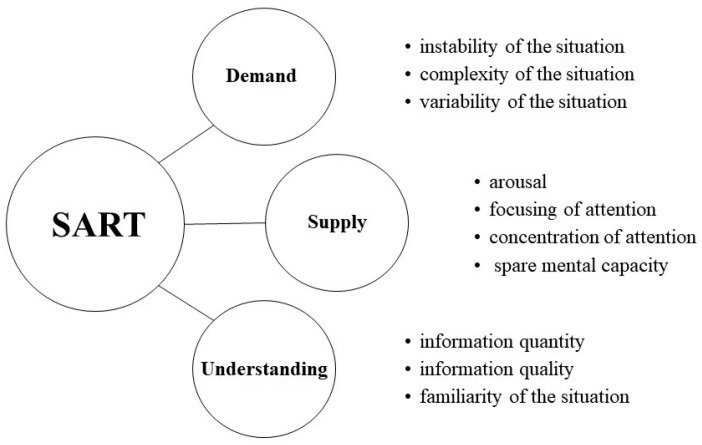
The structure of the situation awareness rating technique.

**Figure 3 behavsci-13-00231-f003:**
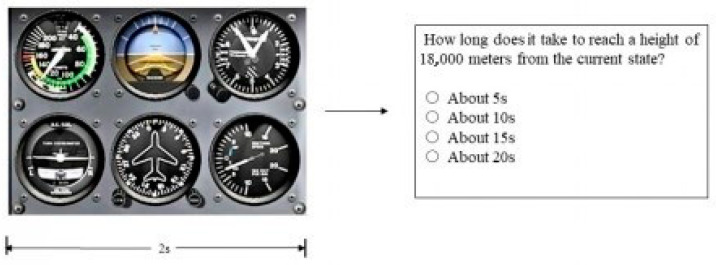
SA task example.

**Figure 4 behavsci-13-00231-f004:**
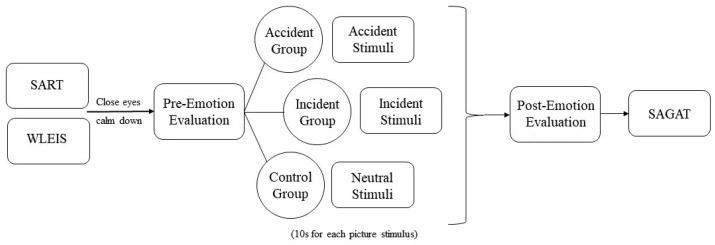
Procedure list.

**Figure 5 behavsci-13-00231-f005:**
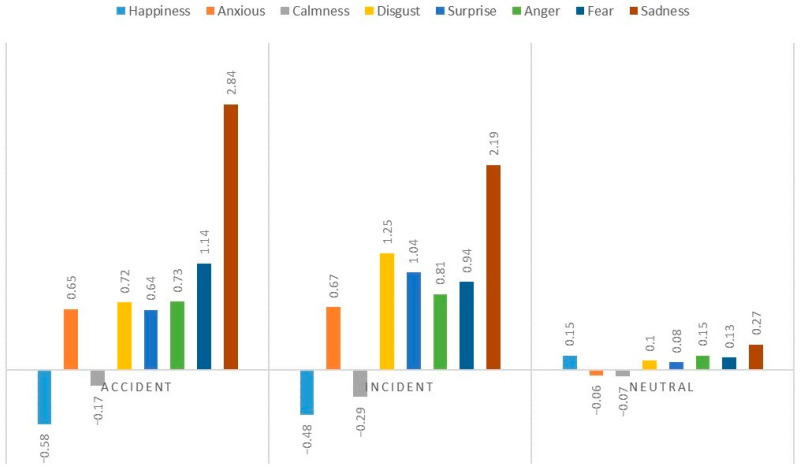
Pilot’s emotional state of various dimensions under different induction conditions.

**Figure 6 behavsci-13-00231-f006:**
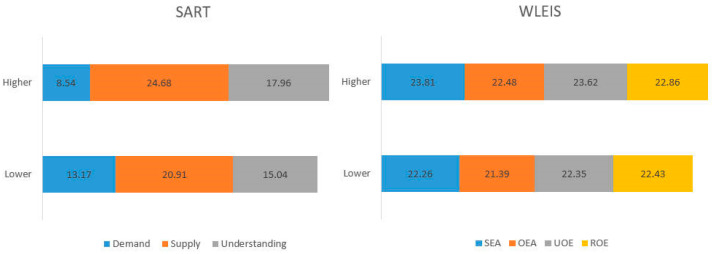
The difference between high and low groups of SART and WLEIS.

**Figure 7 behavsci-13-00231-f007:**
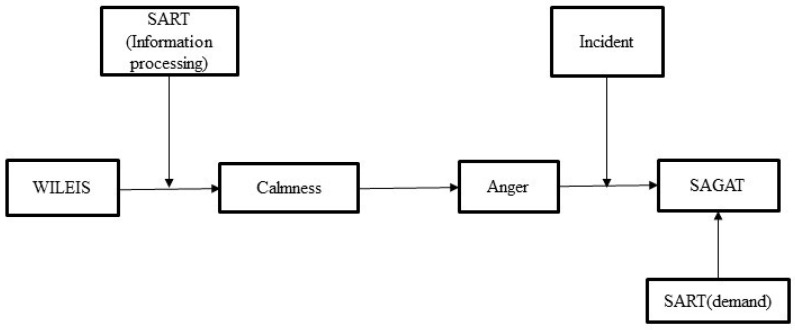
Conditional process model diagram.

**Table 1 behavsci-13-00231-t001:** Effect of Pilot’s Emotional Intelligence on the Rate of Emotional Change After Induction.

SA		Happiness			Anxiety			Calmness			Disgust			Surprise			Anger			Fear			Sadness		
		*F*	*p*	*Eta*	*F*	*p*	*Eta*	*F*	*p*	*Eta*	*F*	*p*	*Eta*	*F*	*p*	*Eta*	*F*	*p*	*Eta*	*F*	*p*	*Eta*	*F*	*p*	*Eta*
High	Accident	6.18	0.03	0.29	3.59	0.08	0.19	1.15	0.30	0.07	2.24	0.16	0.13	0.01	0.91	0.00	0.01	0.93	0.00	1.07	0.32	0.07	1.50	0.24	0.09
	Incident	9.35	0.01	0.38	0.10	0.76	0.01	3.98	0.07	0.21	6.74	0.02	0.31	2.71	0.12	0.15	2.07	0.17	0.12	2.93	0.11	0.16	6.83	0.02	0.31
	WLEIS	0.00	0.96	0.00	3.77	0.07	0.20	7.82	0.01	0.34	0.13	0.73	0.01	0.00	0.99	0.00	0.18	0.68	0.01	4.87	0.04	0.25	7.05	0.02	0.32
	Int 1	0.08	0.78	0.01	2.60	0.13	0.15	4.97	0.04	0.25	0.32	0.58	0.02	0.30	0.59	0.02	0.28	0.61	0.02	5.22	0.04	0.26	6.98	0.02	0.32
	Int 2	1.59	0.23	0.10	6.30	0.02	0.30	0.86	0.37	0.05	13.14	0.00	0.47	2.26	0.15	0.13	2.77	0.12	0.16	0.00	0.99	0.00	0.31	0.59	0.02
Low	Accident	9.63	0.01	0.36	0.01	0.93	0.00	0.03	0.86	0.00	0.46	0.57	0.03	1.02	0.33	0.06	1.26	0.28	0.07	1.28	0.27	0.07	5.01	0.04	0.23
	Incident	3.08	0.10	0.15	0.52	0.48	0.03	0.96	0.34	0.05	3.03	0.10	0.15	1.47	0.24	0.08	1.39	0.26	0.08	0.76	0.40	0.04	0.70	0.41	0.04
	WLEIS	0.10	0.75	0.01	0.91	0.35	0.05	3.53	0.08	0.17	0.42	0.53	0.02	0.34	0.57	0.02	0.03	0.86	0.00	0.47	0.50	0.03	0.00	0.97	0.00
	Int 1	0.01	0.93	0.00	1.03	0.32	0.06	0.01	0.93	0.00	0.08	0.78	0.01	0.01	0.91	0.00	0.04	0.85	0.00	0.33	0.58	0.02	0.02	0.90	0.00
	Int 2	0.02	0.89	0.00	1.81	0.20	0.10	0.04	0.84	0.00	0.18	0.68	0.01	0.11	0.74	0.01	0.00	0.96	0.00	0.09	0.77	0.01	0.01	0.92	0.00

Note: Int 1: Accident × WLEIS; Int2: Incident × WLEIS.

**Table 2 behavsci-13-00231-t002:** The multiple linear regression models of induced condition on SA.

Condition		Items	Unstandardized Coefficients		Standardized Coefficients	*t*	*p*
			Beta	SE	Beta		
Accident	Level1	surprise	0.47	0.14	0.67	3.38	0.00
	Level2	-	-	-	-	-	-
	Level3	Sadness	0.18	0.07	0.55	2.44	0.03
	SAGAT	Sadness	0.34	0.12	0.60	2.77	0.02
Incident	Level1	-	-	-	-	-	-
	Level2	demand	−0.18	0.08	−0.55	2.38	0.03
	Level3	-	-	-	-	-	-
	SAGAT	demand	−0.28	0.12	−0.56	2.41	0.03

**Table 3 behavsci-13-00231-t003:** Moderating effect index.

Regression Equation		*R*	*R* ^2^	*SE*	*F*	*p*	*β*	*se*	*t*	*p*	*95% CI*
Outcome Variable	Predictor Variable										
Calmness		0.66	0.43	0.08	6.63	0.00	-	-	-	-	-
	WLEIS	-	-	-	-	-	−0.18	0.05	−3.52	0.00	[−0.29, −0.07]
	SART (information processing)	-	-	-	-	-	−0.02	0.05	−0.45	0.66	[−0.14, 0.09]
	WLEIS × SART	-	0.19		8.58	-	0.14	0.05	2.93	0.01	[0.04, 0.24]
Anger		0.36	0.13	1.09	4.13	0.05					
	Calmness	-	-	-	-	-	−1.14	0.56	−2.03	0.05	[−2.29, 0.01]
SAGAT		0.63	0.39	2.38	4.04	0.01					
	Anger	-	-	-	-	-	−0.20	0.27	−0.70	0.49	[−0.75, 0.37]
	Incident	-	-	-	-	-	−0.66	0.59	−1.12	0.27	[−1.88, 0.56]
	Anger × Incident	-	0.17	-	7.05	-	−1.40	0.53	−2.65	0.01	[−2.49, −0.31]
	SART (Demand)	-	-	-	-	-	−0.17	0.08	−2.22	0.04	[−0.32, −0.01]

**Table 4 behavsci-13-00231-t004:** Parameter estimates of conditional process model.

Path	Condition	SART	*Effect*	*SE*	*95% CI*
WLEIS → Calmness → Anger → SAGAT	Accident	High	0.02	0.06	[−0.09, 0.16]
		Moderate	0.47	0.08	[−0.10, 0.28]
		Low	0.19	0.20	[−0.22, 0.58]
	Incident	High	−0.04	0.10	[−0.26, 0.12]
		Moderate	0.53	−0.17	[−0.47, −0.00]
		Low	−0.38	0.24	[−0.93, −0.01]

## Data Availability

The datasets analyzed during the current study are not publicly available due to confidentiality agreement, but are available from the corresponding author on reasonable request.

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
