# Peer review of "The Influence of Emotion Induced by Accidents and Incidents on Pilots’ Situation Awareness"

_behavsci, 2023, doi:10.3390/bs13030231_

Round 1

Reviewer 1 Report

The subject of this article is to present the influence of accident and incident stimulus on pilot emotions and Situation Awerness. The article is based on the sense of importance of developing emotion and reaction training for pilots to promote their Sa and performance as well as aviation security. The research was conducted among the jet pilots. The structure and style for article is appropriate. the article is introduced with an abstract, which provided the stance or thesis developed by the article as well as a brief overview of main points. The rationales for the article and for the research it describes are also included. All the methodology procedures are described clearly and comprehensive providing the reader enough information about the primary reaserch. The article is comprehensive of majcor facts related to the topic and presents general and acknowledged views fairly. The list of publication is relevant and update. Overall the article is well written, clear and comprehensive. 

Author Response

We appreciate the reviewer for this kind recommendation. Your comments have given us great encouragement and confidence to improve the article better. Thank you for this valuable feedback.

Reviewer 2 Report

This a solid piece of research on an important subject. The results would benefit from a slight rewrite in the discussion section with clarity and consistency in mind. The theoretical model and its translation to the empirical model might require elaboration. This would be especially useful in regard to the justification for the conditional model you tested.

Author Response

Per your suggestions, we have simplified the content of the introduction section to make it clearer. In the manuscript, we indicated revised text with red font. Thank you very much for your careful review and valuable comments that it’s improved this paper a lot.

Reviewer 3 Report

Brief summary

The objective of the manuscript is to examine the factors associated with aviation accidents. Specifically, it aims to investigate i) how images of aviation accidents and incidents influence pilots' emotions and the moderating role of their emotional intelligence, and ii) how pilots' emotions influence their situational awareness. The study reveals an important interplay of emotional and cognitive factors on situational awareness and provides valuable conclusions for the aviation industry.

Introduction

In general the introduction contains relevant and essential information about the subject of the paper. However, in several places it is not clear how and in what direction the different factors leading to aviation accidents are related. In order to better follow the article, I suggest that the authors describe at the beginning of the introduction how these different factors are interrelated. For example: 1) negative emotions can lead to loss of SA, 2) loss of SA can lead to aviation accidents, 3) aviation accidents can cause negative emotions, 4) which brings us back to point 1. Also, I suggest that the authors describe the moderating effect of emotional intelligence in a bit more detail.

Materials and Methods

The method contains most of the essential information, but could be improved in some respects. It would be necessary to ensure that i) the presentation of the materials is clearly structured (see specific comments), ii) and that the presentation of the materials is consistent with the description of the procedure. For example, it is not clear whether there is a difference between "emotion elicitation" and "experimental group" (accident, incident, neutral) or whether they are one and the same part of the study. I would suggest that the authors consider including a flowchart of the experiment in the article, as it would make the order of the different experimental parts easier to understand. Also, the authors should clearly state how the ethical issues were handled. I am also interested to know if the experimental data/materials will be made publicly available.

Results

The authors used a wide range of statistical methods to test their hypotheses. For the most part, the results are presented in a clear form. However, because certain parts of the method are not entirely clear, some parts of the results are also not entirely understandable. I would suggest the authors first improve the parts of the method where information is missing/unclear (see specific comments), and then consider whether certain changes to the results are also needed. I would also suggest that the authors use figures of the results instead of tables. This would make it easier to understand the relationships between variables and the results in general. I would suggest moving the tables to the supplement.

Discussion

It appears that the authors have done a good job detailing the results obtained in light of their hypotheses and the existing literature. However, given my previous comments, I will be able to provide more thorough feedback once the issues raised above are resolved. In general, I suggest that the authors clarify the structure of the discussion somewhat. For example, they could divide the discussion into subsections with informative subheadings and remove repetitive information.

Specific comments

Introduction

P1, L5 - what kind of “failed decisions”?

P1, L5 - “SA” needs to be defined at the first occurrence.

P1, L19 - “controllable and predictable”: in the following lines the authors write that the pilot’s state (physical and mental) is uncontrollable and unstable —> it seems that the pilot is not completely controllable and predictable as written in L19.

P2, L3 - I propose the authors make a distinction between accidents and incidents already at this point.

P2, L3 - “Besides, the severity of accidents generates varying emotional influences on pilots.” This sentence needs to be clarified (try explaining what you mean by “emotional influences”).

P2, L6 - “influence of SA on overall cognitive abilities”. Is it true that SA influences cognitive abilities or is it the other way around?

P2, L8 - “induction of accidents and incidents to pilots’ emotions” Not clear; if I understand correctly, what you are trying to say is that the accidents and incidents induce pilots’ emotions.

P2, L10 - Which factors exactly?

P4, L10 - “Recently, there is no empirical study on SA influence of the emotional state on SA.” - clarify this sentence (it is not entirely clear what you are trying to state).

P4, L11 - What exactly is this second hypothesis?

P5, L1 - “the importance of emotions to SA and pilot emotions.” Clarify the sentence.

P5, L20 - “offer the role of pilot emotional intelligence.” The role of emotional intelligence has to be more thoroughly explained/presented in the introduction.

P6, L1 - “Aviation accidents can be induced by pilots’ negative emotions or loss of SA, and possibly impact pilots’ emotions.” I think this sentence somehow presents the whole idea of this paper, which is that 1) negative emotions can result in the loss of SA, 2) loss of SA can induce aviation accidents, 3) aviation accidents can induce negative emotions, 4) which leads us back to the point 1. Although the above sentence is not written in such “point-point” way, I would recommend that this idea is elaborated at the beginning of the introduction, as it would make it easier for the readers to follow the paper. In addition, you should also add to this the influence of emotional intelligence, which is a mediating factor in point 2.

P6, H2 - “accidents and incidents’ influence on pilot emotions is subject to the regulating effects of individual emotional intelligence and SA level” . This hypothesis is not clear. Is it that SA level and emotional intelligence mediate the influence of accidents and incident’s on pilot emotions? Or is it that emotional intelligence mediates the effects of accidents and incidents’s on pilot emotions and SA level? In the case of the former, what do you mean by “SA level” - is it the “baseline SA level” (before induction)? This needs to be clarified

Materials and Methods

P1, L3 - Please explain why you included 45 participants. Is it that there are about 15 participants in each experimental group, which matches the number of participants in previous studies?

P2, L1 - Emotion elicitation: it is not clear whether each type of stimuli were presented to separate experimental group (accidents only to accident group…). Is “emotion elicitation” the same as “induction group”? This needs to be clarified.

P3, L1 - “Professional pilots evaluated the emotions of each picture in every group.” Was this a part of the pilot study, or did the study participants evaluated these pictures? This needs to be clarified.

P3, L2 - “The emotional dimension of each picture was rated.” This needs to be explained - what dimension/dimensions?

P3, L3 - “The evaluation results showed that the pilots' evaluation of the induced pictures was per the picture groupings.” Not clear.

P4 - Specify dimensions and the points on Likert scale (what is 1, what is 7). In what order were the stimuli shown? Was it the same for all participants? Which software was used to present the stimuli?

P4 and P5 (”Emotional rating) - These two paragraphs seems to present both the emotion induction by pictures and the assessment of current emotional state. I suggest that the authors clearly divide this into two paragraphs (one for rating the emotions induced by pictures and one related to current emotional state).

P6 - Explain why two different SA measurement methods were used. While you describe SAGAT method in detail, it would also be valuable to describe the SART method in more detail (not only which dimensions participants evaluate; but also what this is based on - what are the participants observing in this task). What score related to SART was used in the analysis?

Procedure: “Between each level of the experiment, the mood-induced reinforcement of the pilot was done (10 stimulus pictures) before entering the next level of testing.” This part needs to be explained in more detail in the materials. Is this the same as “emotion elicitation” and/or experimental condition (neutral, accident, incident)? Also, the authors state that mood-induction was performed between each level of the experiment - what exactly does that mean; how many times and between which exact phases?

Results

P1 - “Variance test results found a significant differences among three evoked conditions of happiness …” Please explain the direction of the differences (in which condition was the change rate bigger/smaller)? Please take this comment into account also in other parts when you talk about the significant differences.

P2 - Please check if the subsection’s title is correct (it seems almost the same as the next one).

P2, L1 - The four scoring dimensions of SART need to be mentioned in the methods part.

P3, L1 - “Analysis of influence of main factors conducted on four dimensions” - four dimension on what? WLEIS?

P4: “Comparison of happiness in neutral-induced control group shown a significant emotion change in pilots in the high-score group under both accident and incident conditions (R 2= 0.47).” This part is not clear, as it is not clearly stated in the method what is the difference between experimental groups and emotion elicitation conditions.

Results: “The effects of emotional quotient and SART on SAGAT after induction” - This part of analysis is not clear - how exactly were the models built (in what order); please explain the levels.

Discussion

P1, L3 - “positive control group” - Do you mean neutral control group?

P1, P4 - “Unlike Hypothesis 1, the direct discrepancy of induction emotions under two conditions is less significant.” Please remind us what is the Hypothesis 1. Please take this comment into account elsewhere.

Author Response

Thank you very much for your careful review and valuable comments that it’s improved this paper a lot. We studied the comments carefully and made revisions that we hope will be met with your approval. In the manuscript, we indicated revised text with red font. Please see the attachment, in which there are the corrections in the manuscript and the responses to your comments.
